# Association of Prenatal Polycyclic Aromatic Hydrocarbons Exposure, DNA Hydroxymethylation, and Neurodevelopment at 0 and 2 Years of Age

**DOI:** 10.3390/toxics13090726

**Published:** 2025-08-29

**Authors:** Jinyu Li, Xiaomin Cao, Chengjuan Liu, Lin Cheng, Qiao Niu, Jisheng Nie

**Affiliations:** 1Section of Occupational Medicine, Department of Special Medicine, Shanxi Medical University, Taiyuan 030001, China; captain@sxmu.edu.cn; 2Department of Occupational and Environmental Health, School of Public Health, Shanxi Medical University, Xinjiannan Road 56, Taiyuan 030001, China; 201700510668@b.sxmu.edu.cn (X.C.); 18865675969@163.com (C.L.); clasxjysys_1011@163.com (L.C.); niuqiao55@163.com (Q.N.)

**Keywords:** polycyclic aromatic hydrocarbons, 5-hydroxymethylcytosine, *BDNF*, *MeCP2*, neurobehavioral development, prenatal exposure

## Abstract

Maternal exposure to polycyclic aromatic hydrocarbons (PAHs) during pregnancy may have effects on the offspring epigenome. And the change in onset epigenome may be associated with children’s neurodevelopment. The current study investigated the relationship between 5-hydroxymethylcytosine (5-hmC) levels in cord blood and PAH metabolites in maternal urine at delivery and children’s neurodevelopment at birth and at age 2. We enrolled 400 pregnant women and their newborns and collected their biological samples after obtaining written informed consent. Enzyme linked immunosorbent assay kits and Chromatin immunoprecipitation kits were used to assess the DNA hydroxymethylation level in cord blood. We observed that 1-hydroxypyrene (1-OHPyr) was inversely associated with gesell developmental scale scores, positively associated with global DNA 5-hmC levels, and associated with decreased 5-hmC levels of the brain-derived neurotrophic factor (*BDNF*) and methyl CpG binding protein 2 (*MeCP2*) gene promoter. In addition, the 5-hmC levels of the *BDNF* and *MeCP2* gene promoters were associated with motor scores. The global DNA 5-hmC was inversely associated with motor scores. Mediation analysis showed mediation effects between 1-OHPyr and motor scores by 5-hmC. The global DNA 5-hmC and *MeCP2* and *BDNF* gene promoter 5-hmC contributed 28.51%, 27.29%, and 18.98% of the effect on motor scores changes related to 1-OHPyr. The study results suggested that 5-hmC can be a potential mechanism between prenatal PAH exposure and children’s neurodevelopment at age 2 and provide a better understanding of the role of hydroxymethylation in neurodevelopment.

## 1. Introduction

Polycyclic aromatic hydrocarbons (PAHs) belong to a large family of widespread environmental pollutants produced from incomplete combustion of fossil fuels and carbon-containing materials [1,2]. Epidemiologic evidence has revealed that prenatal PAH exposure is associated with adverse neurodevelopmental outcomes. Biomarkers predicting adverse neurodevelopmental outcomes are vital for identifying children who are affected by prenatal PAH exposure before it becomes clinically evident.

Epigenome changes often serve to discover biomarkers of disease and to understand biologic processes. Human brain development is characterized by coordinated changes in gene expression mediated by a complex interaction between transcription factors and epigenetic processes. Studies have shown that epigenetic modification can play an important role in neurodevelopment [3,4,5]. One of the most widely studied epigenetic modifications is DNA methylation at the 5′ position of cytosine nucleotides, also referred to as 5-methylcytosine (5-mC) [3]. It has been demonstrated that 5-mC can undergo oxidation by the ten-eleven translocation (TET) family of enzymes to form 5-hydroxymethylcytosine (5-hmC) [6]. Regulation of gene expression by 5-hmC has been shown to be independent of 5-mC and plays a crucial role during embryogenesis, particularly in neurogenesis [7,8]. We speculate that the hydroxymethylation mark, 5-hmC, may be a potentially important biomarker in neurodevelopment because the hydroxymethyl group on the cytosine can possibly switch a gene on and off. Of note, 5-hmC is present at relatively high levels in the central nervous system, and is particularly enriched in the vicinity of genes with synapse-related functions [9,10,11]. Previous studies have reported that 5-hmC may be especially important in central nervous system development and suggest that changes in global DNA hydroxymethylation may regulate brain development [3,5,12]. The exertion of any effect depends on a specific material basis, and changes in 5-hmC of neurodevelopment-related genes may be an important biomarker. The methyl CpG binding protein 2 (*MeCP2*) gene and brain-derived neurotrophic factor (*BDNF*) gene are two important genes involved in neurodevelopmental processes [13,14]. The protein encoded by the *MeCP2* gene is a transcriptional regulator that can regulate the expression of multiple genes in the brain [15]. It functions as a transcriptional activator and repressor. Studies have shown that MeCP2 is closely related to neurodevelopment [16]. BDNF is the most abundant neurotrophic factor in the brain, which can increase synaptic plasticity and promote neurogenesis [17]. Studies show that the effect of PAH on neurodevelopment is partly attributed to the decrease in BDNF level in cord blood plasma [18]. The 5-hmC levels of the *MeCP2* and *BDNF* genes may affect the neurodevelopmental level of the offspring. Therefore, identifying whether 5-hmC changes at birth are associated with prenatal exposure to PAH will help elucidate their potential role as biomarkers of exposure or disease susceptibility.

In addition, epigenetic modification is highly vulnerable to environmental exposure [19]. At present, a small number of studies have successively reported the relationship between 5-hmC and environmental chemicals and neurobehavioral development. Human studies suggest that exposure to ambient particulate matter affects the 5-hmC level in peripheral blood over time [20]. However, no study has explored the impact of prenatal PAH exposure on global DNA and specific gene promoter 5-hmC levels. Additionally, the role of 5-hmC between prenatal PAH exposure and child neurobehavioral developmental outcomes is still unclear. Using an epigenome-kit approach, we aimed to identify changes in DNA hydroxymethylation in neonates associated with prenatal maternal exposure to PAH. We hypothesized that prenatal exposure to PAH would lead to differences in DNA hydroxymethylation in cord blood that would persist into childhood. The purpose of this study was to investigate (1) whether PAH exposure is associated with altered global DNA 5-hmC and *MeCP2* and *BDNF* gene promoter 5-hmC and (2) whether changes in 5-hmC mediate the associations between prenatal PAH exposure and children’s neurobehavioral developmental outcomes.

## 2. Materials and Methods

### 2.1. Study Population

Pregnant women who waited for delivery during the third trimester of pregnancy (≥35 weeks) in the Sixth Hospital of Shanxi Medical University and the Eighth People’s Hospital of Taiyuan were invited to participate in the Taiyuan Mother and Child Cohort Study. Eligibility criteria included the following: eligible women had resided in Taiyuan city for at least 1 year, were ≥18 years old, were non-smokers, had no family history of neurologic diseases and chronic diseases, and had a single gestational viable fetus. Before entering the study, informed consent was obtained from mothers. If pregnant women met the inclusion criteria and signed informed consent, we collected maternal urine before delivery. Umbilical cord blood was collected after delivery. A total of 400 mother–newborn pairs were included in the cohort study. The present study obtained approval from the Ethical Committee of Shanxi Medical University Institutional Review Board (No. 2016LL087). Two years later, 221 mother–child pairs were included in the cohort.

### 2.2. Personal Interview

A 45 min questionnaire was conducted within three days after delivery to collect parental demographics, socioeconomic status, lifetime residential history, environmental exposure history, pregnancy history, and medical history. During follow-up, questionnaires with several scales were completed by professional doctors. There were 221 mother–child pairs with 2-year-old children included in the study. The Institutional Review Boards of Shanxi Medical University approved the follow-up study, and informed consent was obtained prior to birth and when the children were 2 years old.

### 2.3. Urine Collection and PAH Metabolites Analysis

A first morning urine sample was collected with a urine container from all participants when they were admitted to the hospital and stored at −80 °C until analysis. Each urine sample was measured for a suite of PAH metabolites: 2-hydroxynaphthalene (2-OHNap), 1-hydroxynaphthalene (1-OHNap), 3-hydroxyfluorene (3-OHFlu), 2-hydroxyfluorene (2-OHFlu), 9-hydroxyphenanthrene (9-OHPhe), 2-hydroxyphenanthrene (2-OHPhe), 1-hydroxyphenanthrene (1-OHPhe), 1-hydroxypyrene (1-OHPyr), 3-hydroxychrysene (3-OHChr), 6-hydroxychrysene (6-OHChr), and 9-hydroxybenzo[a]pyrene (9-OHBap). We used high-performance liquid chromatography with tandem mass spectrometry (HPLC-MS/MS (Shimadzu, Kyoto, Japan)) to measure the above eleven PAH metabolites as described previously [21].

### 2.4. Umbilical Cord Collection and BPDE Analysis

A total of 30–50 mL umbilical cord blood was collected after delivery into whole blood tubes (BD, Franklin Lakes, NJ, USA). We applied the high-performance liquid chromatography (HPLC)/fluorescence method to detect the BPDE ((+)benzo[a]pyrene-7,8-dihydrodiol-9,10-epoxide) in cord blood white blood cells as described previously [22]. The linearity (expressed as R^2^) was 0.98, the mean coefficient of variation on different days was 12%, and the limit of detection (LOD) was 0.25 adducts/10^8^ nucleotides.

### 2.5. Global DNA 5-hmC in Cord Blood Analysis

Umbilical cord blood was collected after delivery into whole blood tubes (BD, Franklin Lakes, NJ, USA). Genomic DNA was isolated from the buffy coat of cord blood using a universal genomic DNA kit (Cwbiotech, Beijing, China). The levels of global DNA 5-hmC were measured by an ELISA kit according to the manufacturer’s protocols, as described previously [20]. In brief, the standard 5-hmC was used to generate a fresh five-point standard curve (range: 0–0.55%). Then, according to the protocols, 100 ng of genomic DNA in each reaction was used to measure the global 5-hmC level by the Quest 5-hmC ^TM^ DNA ELISA Kit (Zymo Research, Irvine, CA, USA). All experiments were run in triplicate. DNA samples were randomized across plates to limit potential bias from plate effects. The kit is highly sensitive and specific to the 5-hmC modification of concern, with no cross-reactivity for other modifications. The R^2^ of the standard curve was 0.99. The within- and between-assay coefficients of variation were 3.9% and 10.8%, respectively, for 5-hmC. The 5-hmC detection limit per 100 ng of input DNA was 0.02%.

### 2.6. BDNF and MeCP2 Gene Promoter Hydroxymethylated DNA Immunoprecipitation Analysis

The 5-hmC levels in the promoter regions of the *BDNF* and *MeCP2* genes were determined according to the protocol of the EpiQuik^TM^ Hydroxymethylated DNA Immunoprecipitation Kit (Epigentek, Farmingdale, NY, USA). Briefly, genomic DNA was sheared to generate fragments of approximately 100–600 bp by sonication. Each sample of the sonicated DNA was immunoprecipitated with the human monoclonal 5-hmC antibody or with normal IgG. DNA from the antibody-bound fractions was purified with proteinase K in DNA isolation buffer. The purified immunoprecipitated DNA and input DNA were subsequently amplified with q-PCR, and the percent enrichment was calculated by the formula: 100 × 2 ^(adjusted input (Ct)—input (Ct))^. The primers used for the determination of the 5-hmC levels of *BDNF* and *MeCP2* came from a previous study [23] and Appendix A.

### 2.7. Outcomes

When newborns were three days old, newborns were administered the neonatal behavioral neurological assessment (NBNA) test. To maximize reliable assessment and valid interpretation, testing was conducted by pediatricians in the study from delivery hospitals who were certified in the NBNA, minimizing both inter- and intra-examiner variability. The detailed information was described in our previous study [21].

The Chinese version of the standardized Gesell Developmental Scales (GDSs) for 0- to 3-year-old children, adapted to the Chinese population by the Chinese Pediatric Association, was administered to children in this cohort at 2 years of age [24,25]. Each child was assigned a development quotient (DQ) in each of four areas: motor, adaptive, language, and social. To minimize inter-examiner and intra-examiner variability, two trained and certified physicians conducted all testing. In the present study, the neurobehavioral developmental score served as a continuous variable for analysis.

### 2.8. Covariates

Covariates were considered in this study, including the mother’s age, mother’s education (middle school and below vs. high school vs. college and above), pre-pregnant BMI, household income (≤CNY 30,000, CNY 30,000–50,000, and ≥CNY 50,000 per year), parity (single vs. multiply), passive smoking (yes vs. no), child’s sex (female vs. male), gestational age, and cord blood lead. Passive smoking was defined as a pregnant woman who was exposed to environmental cigarette smoke >15 min per day at work and/or at home [26]. The GDS scores were additionally adjusted for the level of PAH metabolites at age 2. Heavy metals except lead, and co-pollutants were not considered as environmental confounders in this study.

### 2.9. Statistical Analysis

The basic characteristics of the mother–newborn pairs were shown as the mean (SD) or frequency (proportion). Comparison of two groups was performed with one-way ANOVA or chi-square test. The distribution of PAH metabolites levels was presented as the median and percentiles (25th and 75th). To ensure statistical efficiency, a sufficient sample size is a prerequisite for analyzing the relationship between PAH metabolites and outcomes. Therefore, PAH metabolites with a detection rate of less than 50% were excluded from the analysis in this study. The detection rate of 3-OHChr, 6-OHChr, and 9-OHBap in maternal urine was less than 50%, and we therefore did not include the three urinary PAH metabolites in the final analysis. Samples with PAH metabolites and 5-hmC levels that were less than the limit of detection were given a value of 1/2 the limit of detection [21]. To normalize the right-skewed distributions, PAH metabolites levels and 5-hmC levels were log transformed, while the neurobehavioral developmental test scores were modeled in their original scale. PAH metabolites levels were also categorized into tertiles, and the test for trends was conducted for ordinal PAH metabolites categories in general linear models using integer values (1–3). For models with significant statistical association estimates, we further used restricted cubic spline models to assess the shapes of the dose–response association.

Mediation analysis investigates the mechanisms that underlie an observed relationship between an exposure variable and an outcome variable and examines how they relate to a third intermediate variable, the mediator [27]. The mediator variable then serves to clarify the nature of the relationship between the exposure and the outcome variable. Consensus has now been reached that the mediator needs to satisfy the following criteria [28]: (i) a change in the levels of the exposure variable significantly affects the changes in the mediator; (ii) there is a significant relationship between the mediator and the outcome; and (iii) a change in the levels of the exposure variable significantly affects the changes in the outcome. A mediation analysis was conducted to test the role of 5-hmC in the associations between PAH metabolites and neurodevelopmental outcomes because PAH metabolites were associated with 5-hmC and neurodevelopmental outcomes and 5-hmC was associated with neurodevelopmental outcomes. All results were considered statistically significant if the binary *p*-value was less than 0.05. All statistical tests were performed using SAS 9.4 (SAS Institute, Inc., Cary, NC, USA).

## 3. Results

### 3.1. Characteristics of the Participants

The characteristics of the study population are shown in Table 1. There were no statistically significant differences between participants and non-participants in demographic characteristics, except for maternal educational status, household income, and children’s birth weight and birth length. The proportion of college and above in this study (56.6%) was higher than that in the parent study (46.2%), which revealed that the follow-up compliance of subjects with higher education is better than that of subjects with lower education levels. The mean values of motor, adaptive, language and social scores were 111.6, 110.5, 108.8, and 111.7, respectively.

### 3.2. PAH Metabolites Levels

The distributions of maternal urinary PAH metabolites and cord blood BPDE levels are presented in Table 2. Generally, the measured PAH metabolites were readily detectable in urine of pregnant women, expect for 3-OHChr, 6-OHChr, and 9-OHBap. The median (P50) level of urinary 2-OHPhe was the highest (0.133 ng/mL), followed by urinary 1-OHPyr (0.090 ng/mL). The median level of cord blood BPDE was 5.2 × 10^8^ nucleotides.

### 3.3. PAH Metabolites and Neurobehavioral Developmental Scores

General linear analysis and tests for trend results showed significant relationships between higher 1-OHPyr and lower GDS scores, especially motor score (*p* for trend < 0.01) (Figure 1). Furthermore, the dose–response association of maternal urinary 1-OHPyr with decreased GDS scores was confirmed in the restricted cubic spline models (*p* < 0.05) (Figure 2). The motor score mainly reflected the development of gross and fine motor skills in children. Low motor scores were associated with a higher risk of developing autism spectrum disorder. Compared to the 1st tertile, the 3rd tertile group showed a decrease of 10.3 points in motor scores and a decrease of 9.2% compared to the average level, indicating an increased risk of developing autism and motor delay.

### 3.4. PAH Metabolites and Cord Blood 5-hmC Levels

The associations between PAH metabolites and 5-hmC levels are shown in Figure 3. We observed that maternal urinary 1-OHPyr was associated with increased trends for global DNA 5-hmC and decreased trends for *BDNF* and *MeCP2* gene promoter 5-hmC levels (*p* for trend < 0.05). Furthermore, the dose–response associations of maternal urinary 1-OHPyr with increased global DNA 5-hmC (*p* < 0.01) and with decreased *MeCP2* and *BDNF* gene promoter 5-hmC (*p* < 0.05) were confirmed in the restricted cubic spline models (Figure 4).

### 3.5. The Associations Between 5-hmC and Neurodevelopmental Indexes

Table 3 shows the associations between 5-hmC and neurobehavioral developmental scores. The global DNA 5-hmC was inversely associated with the NBNA and GDS scores (*p* < 0.05). The level of *MeCP2* gene promoter was positively associated with the NBNA and GDS scores (*p* < 0.05). The *BDNF* gene promoter was only positively associated with increased motor scores (β = 7.77, 95% CI: 3.45, 12.09).

### 3.6. Mediation Analyses

Because maternal urinary 1-OHPyr was associated with GDS scores and 5-hmC levels, and the 5-hmC levels were associated with neurodevelopmental scores, we conducted the mediation analysis to assess whether 5-hmC could be a mediator of the association between maternal urinary 1-OHPyr and GDS scores after adjusting for potential confounders. We found significant mediation effects (indirect effects) between maternal urinary 1-OHPyr and motor scores for global DNA 5-hmC and *MeCP2* and *BDNF* gene promoter 5-hmC (*p* for mediator < 0.05). The global DNA 5-hmC and *MeCP2* and *BDNF* gene promoter 5-hmC could explain 28.51%, 27.29%, and 18.98%, respectively, of the effect of 2-year-old children’s motor scores changes related to 1-OHPyr. The direct effects of 1-OHPyr on the motor scores were 71.49%, 72.71%, and 81.02%, respectively. But, the direct effect of 1-OHPyr on the adaptive scores, language score, and social score was not significant, indicating that 5-hmC played a complete mediating effect between 1-OHPyr and these three scores, reflecting that the impact of 1-OHPyr on adaptive scores, language score, and social score was achieved through its influence on 5-hmC (Table 4).

### 3.7. Sensitivity Analyses

Because the detection rates of 3-OHChr, 6-OHChr, and 9-OHBap were less than 50%, they were directly excluded from the analysis in this study. These metabolites may serve as confounding factors affecting the existing conclusions of this study. Therefore, we included them as covariates in the linear regression model (Appendix A). The results showed that urinary 1-OHPyr was negatively associated with GDS scores (all *p* < 0.05), and urinary 1-OHPyr was associated with increased global DNA 5-hmC and decreased *BDNF* and *MeCP2* gene promoter 5-hmC levels (all *p* < 0.05). These were consistent with the findings of this study, indicating that the relationship between 1-OHPyr and outcomes and biomarkers is not affected by these three metabolites.

## 4. Discussion

To our knowledge, the present study is the first report of the relationship among maternal exposure to PAH, the levels of global DNA 5-hmC, and the specific gene promoter 5-hmC in cord blood and offspring neurodevelopmental scores at birth and 2 years old. In this study, maternal urinary 1-OHPyr was associated with lower neurodevelopmental scores in children ages 0–2 years. A significant inverse association between 1-OHPyr and the 5-hmC levels of the *BDNF* and *MeCP2* gene promoters was also observed. 1-OHPyr was positively associated with global DNA 5-hmC levels. A mediation analysis indicated that 5-hmC served as a mediating role between 1-OHPyr and GDS scores. This suggests that PAH exert adverse neurodevelopmental effects through changes in DNA hydroxymethylation modification and that 5-hmC can serve as an early predictive biomarker for neurodevelopment.

The results of this study showed that maternal urinary specific PAH metabolites, especially 1-OHPyr, was inversely associated with neurodevelopmental scores at 0–2 years of age. At present, 1-OHPyr is the most extensively studied PAH hydroxyl metabolite and is a reliable biomarker for determining the health effects of human exposure to PAH, which can come from various sources including air pollution, diet, and tobacco [29]. There were numerous urinary PAHs metabolites that have been identified in humans. Among these metabolites, 1-OHPyr has been widely applied as an internal exposure indicator of PAH. Wu et al. [30] found significantly higher levels of 1-OHPyr and 3-hydroxybenzo[a]pyrene in coke-oven workers compared to referents, suggesting their potential as biomarkers for benzo[a]pyrene exposure. Numerous studies have reported a negative correlation between 1-OHPyr and neurodevelopmental indicators [31], which is consistent with our findings.

Several epidemiological studies and animal studies have suggested that prenatal exposure to PAH can lead to adverse effects on neurodevelopmental outcomes [32,33]. This is generally consistent with our findings from this study, measured by urinary PAH hydroxyl-metabolites and cord blood BPDE. What should be noted is whether the levels of PAH hydroxylmetabolites in pregnant women urine are influenced by the fetus. Cathey et al. [34] reported that PAH metabolites have been associated with changes in hormone levels, including reproductive and thyroid hormones, with some links varying based on fetal sex. As an independent living body, the fetus also possesses certain metabolic and detoxification capabilities, indicating that in cross-sectional studies, care should be taken in interpreting causal relationships.

This study is the first to report the levels of global DNA 5-hmC and specific gene promoter 5-hmC in cord blood. Some studies have reported the level of global DNA 5-hmC in adult blood. The median level of global DNA 5-hmC (0.05%) in our study was lower than those reported in adults from the United States (0.09%) [35] and Beijing (0.08%) [20]. In a study by Figueroa-Romero et al., the mean 5-hmC in blood samples was 0.03% [36]. These results indicated regional and population differences in the global DNA 5-hmC levels. Further research needs to be performed in multiple regions and centers.

Several studies have revealed that 5-hmC may respond to environmental exposure, although the results of these studies were inconsistent. In the present study, we found that maternal urinary 1-OHPyr was associated with increased global DNA 5-hmC in cord blood. The positive correlation between PAH exposure and global DNA 5-hmC levels was in line with 5-hmC biology. PAH exposure results in the generation of reactive oxygen species [37,38], which are chemically reactive molecules that can induce the oxidation of 5-mC into 5-hmC [39]. Our findings were also consistent with recent results by Coulter et al. [40], who showed that hydroquinone exposure leads to active DNA demethylation in HEK293 cells in a mechanism involving increased reactive oxygen species. Chia et al. have also proposed that reactive oxygen species affect 5-hmC patterns via metabolic alterations influencing the tricarboxylic acid cycle and thereby activating ten-eleven translocation (TET) and other chromatic modifying proteins [39]. However, in some studies, there were opposite results. A study based on a US pre-birth cohort reported that maternal mercury exposure is associated with lower 5-hmC levels in cord blood, and this association could persist in early childhood [41]. A study conducted in a Chinese population revealed an increase in hydroxymethylation with elevated exposure to particulate matter ≤ 10 μm [20]. A mouse whole-embryo culture model revealed that decreased levels of the percentage of 5-hmC and increased levels of reactive oxygen species were found in mouse embryos treated with benzo[a]pyrene [42]. We speculated that these discrepancies may be due to differences in the time window for examining 5-hmC because 5-hmC is an intermediate in DNA demethylation. These differences suggest that oxidative stress or antioxidant depletion might be involved in the observed association between PAH exposure and a higher global DNA 5-hmC level. Future studies on global DNA 5-hmC should take the time window into consideration. The inconsistencies in these data may also be due in part to differences in population characteristics, tissues, or the methodologies (ELISA or mass spectrometry) chosen for detecting 5-hmC.

To date, the biological mechanism by which PAH exposure causes adverse neurodevelopment remains unclear. DNA hydroxymethylation plays a key role in maintaining genomic stability and the expression of genes, and epigenetic modification is susceptible to environmental factors. Epigenetic deregulation has also been viewed as an attractive intermediate mechanism for poor neurodevelopment in association with PAH exposure. Thus, in this study, DNA hydroxymethylation is hypothesized to be one of the mechanisms by which PAH exposure in pregnancy may be associated with adverse neurodevelopmental outcomes in childhood. Few studies have examined the changes in hydroxymethylation with regard to neurodevelopmental disorders caused by prenatal PAH exposure. In this study, we not only detected global levels of 5-hmC but also evaluated localized hydroxymethylation of the *BDNF* and *MeCP2* gene promoters. We found that global DNA 5-hmC was associated with lower NBNA and GDS scores. The 5-hmC level of *MeCP2* gene was positively associated with NBNA and GDS scores. The 5-hmC level of *BDNF* gene was positively related to motor score.

The present study is the first report to investigate these relationships among PAH exposure, specific gene 5-hmC levels, and neurodevelopment. Our results identified that *BDNF* gene promoter 5-hmC levels were associated with higher neurobehavioral scores. Brain development depends on protein suppression related to neurodevelopmental genes. In theory, 5-hmC affects the switching on and off of genes. High 5-hmC means higher expression activity of genes related to neurodevelopment. Previous studies have shown that the loss of Tet2 and 5-hmC in the aged hippocampus is associated with regenerative decline in animal experiments [43]. A study also reported that decreased 5-hmC levels are associated with reduced striatal A2AR levels in Huntington’s disease [44]. A previous study identified widespread changes in 5-hmC occurring during human brain development [11]. This is consistent with our conclusion that the 5-hmC level of the *BDNF* gene promoter is positively associated with higher motor scores, mediates the associations of PAH with neurodevelopmental indexes, and partly explains the effect of PAH on motor scores. Neural intelligence development often depends on the expression of corresponding proteins or peptides, such as BDNF protein level, whose high expression is conducive to nerve development, while low expression is often associated with mental retardation. Studies have shown that PAH exposure during pregnancy is inversely related to cord blood plasma BDNF levels, and it has been found that the relationship between PAH exposure during pregnancy and offspring neurodevelopmental delay can be explained by lower cord blood plasma BDNF levels [18]. We speculate that this may be because PAH are metabolized to form reactive epoxides, which covalently bind to DNA, thereby interfering with the hydroxymethylation modification of the *BDNF* gene, further affecting the expression of related genes and leading to neurodevelopmental disorders in offspring. This hypothesis is basically consistent with the conclusions obtained in this study: PAH causing 5-hmC changes in the *BDNF* gene can partially explain the effects of PAH exposure during pregnancy on the motor score in 2-year-old children. In short, the 5-hmC change in the neurodevelopmental gene *BDNF* may be one of the mechanisms of PAH causing neurodevelopmental toxicity in pregnancy. In contrast to BDNF, MeCP2 is a transcriptional regulator that can regulate the expression of multiple neurodevelopment-related genes in the brain and serve as a transcriptional activator and repressor. This is consistent with our results that *MeCP2* was positively associated with four neurodevelopmental indexes (NBNA, motor, adaptive, language, and social). Except for the motor score, mediation analysis indicated that the indirect effect of 5-hmC of *MeCP2* and global DNA on neurodevelopment indexes has statistical significance, although the direct effect was not significant in adaptive, language, and social scores. The results suggest that 1-OHPyr completely affects neurodevelopmental indexes (adaptive, language, and social) through changes in the 5-hmC levels of *MeCP2* and global DNA, that is, the 5-hmC of *MeCP2* and global DNA represents the only path between 1-OHPyr and these neurodevelopmental indexes. Epigenetics marker such as methylation and hydroxylation have been applied in the field of aging research on biological age and biological clock [45], which highlight the value of epigenetic modifications as biomarkers. Epigenetic signatures are sensitive markers of environmental toxicity [46]. This study suggested the role of umbilical cord blood hydroxymethylated 5-hmc level in offspring neurodevelopment induced by PAH exposure, suggesting the predictive value of umbilical cord blood bioinformation in offspring development, which is exactly the content of “DOHaD theory” [47]. From the perspective of health economics, the benefit of umbilical cord blood 5-hmC screening is less than the cost, but for pregnant women with occupational exposure, monitoring umbilical cord blood 5-hmC or DNA damage has a certain value for timely intervention in children’s development.

We have to acknowledge that there were some limitations in the present study. First, although urinary PAH hydroxylmetabolites are markers of ambient PAH exposure, there are no personal external exposure PAH data in this study. Urinary PAH hydroxylmetabolites are influenced by metabolic capacity and individual genetic polymorphism. In addition, we excluded PAH metabolites with a detection rate of less than 50%, which directly masked the effects of these metabolites on neurodevelopment. In order to clarify their impact on the results, we used them as covariates for sensitivity analysis to confirm that these metabolites did not affect the original conclusion. Second, both the prenatal and early postnatal periods are critical windows for neurotoxic substance exposure during neurodevelopment in humans and mice [48,49]. We only measured single-point PAH levels in maternal urine to represent the prenatal exposure of the fetus. However, accumulative studies use a single measurement of PAH to serve as a predictive of long-term exposure [25,50,51]. Third, in this study, there were statistical differences in education level, household income level, and neonatal characteristics between participants, and lost to follow-up individuals, suggesting that this study may be a biased population. These biases objectively exist in the selection process. In order to minimize the impact of these factors on the results, we treated these variables as covariates and found that the results still hold true. Finally, we did not take into account other confounding factors that may affect PAH metabolism and outcomes, such as nutrition and genetic polymorphism, which were identified as genes and lifestyle factors that affect PAH metabolism and neurodevelopment.

## 5. Conclusions

In summary, specific PAH metabolites are inversely associated with neurobehavioral developmental scores and 5-hmC levels. Additionally, 5-hmC mediates the associations of PAH with neurobehavioral developmental indexes and partly explains the effect of PAH on neurobehavioral developmental scores in children aged 0–2 years old.

## Figures and Tables

**Figure 1 toxics-13-00726-f001:**
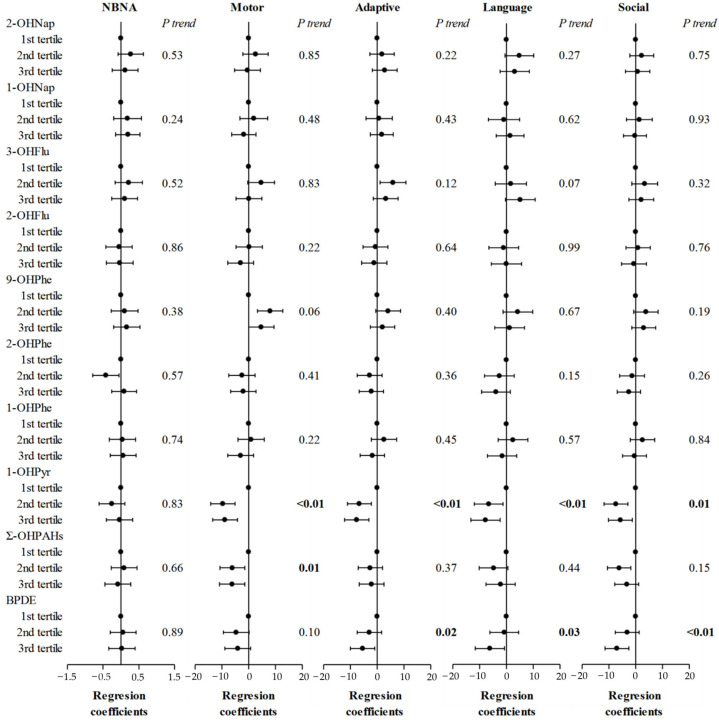
Regression coefficients for neurobehavioral developmental scores associated with PAH metabolites levels after adjusting for mother’s age, mother’s education, pre-pregnant BMI, household income, parity, passive smoking, child’s sex, gestational age, and cord blood lead. The Gesell developmental scale scores were additionally adjusted for the level of PAH metabolites at age 2. Note: NBNA: Neonatal Behavioral Neurological Assessment; Ʃ-OHPAH: sum of 11 individual PAH metabolites; BPDE: (+)benzo[a]pyrene-7,8-dihydrodiol-9,10-epoxide; BMI: body mass index.

**Figure 2 toxics-13-00726-f002:**
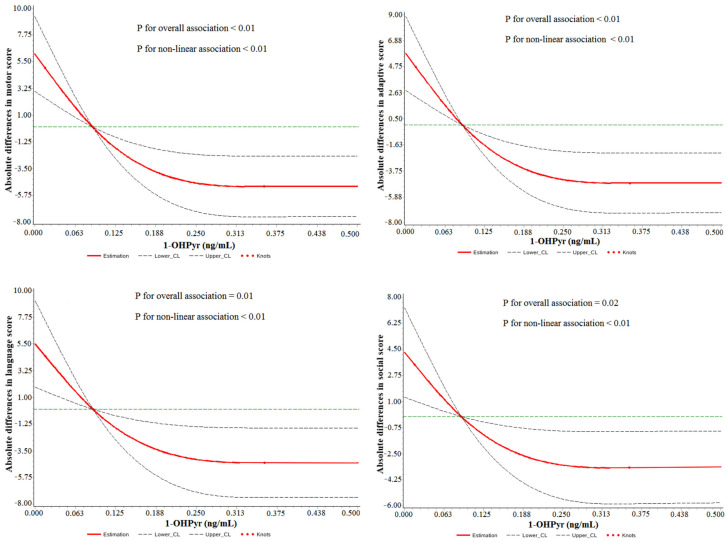
Restricted cubic spline models representing the associations between maternal urinary 1-OHPyr and neurobehavioral developmental scores, adjusted for mother’s age, mother’s education, pre-pregnant BMI, household income, parity, passive smoking, child’s sex, gestational age, cord blood lead, and the level of PAH metabolites at age 2. The reference group is 0.090 ng/mL; dashed lines represent the 95% CI; the green line represent the reference line with a vertical axis of 0; the red knots represent the 1-OHPyr concentrations at the 10th, 50th, and 90th percentiles, respectively. Note: BMI: body mass index.

**Figure 3 toxics-13-00726-f003:**
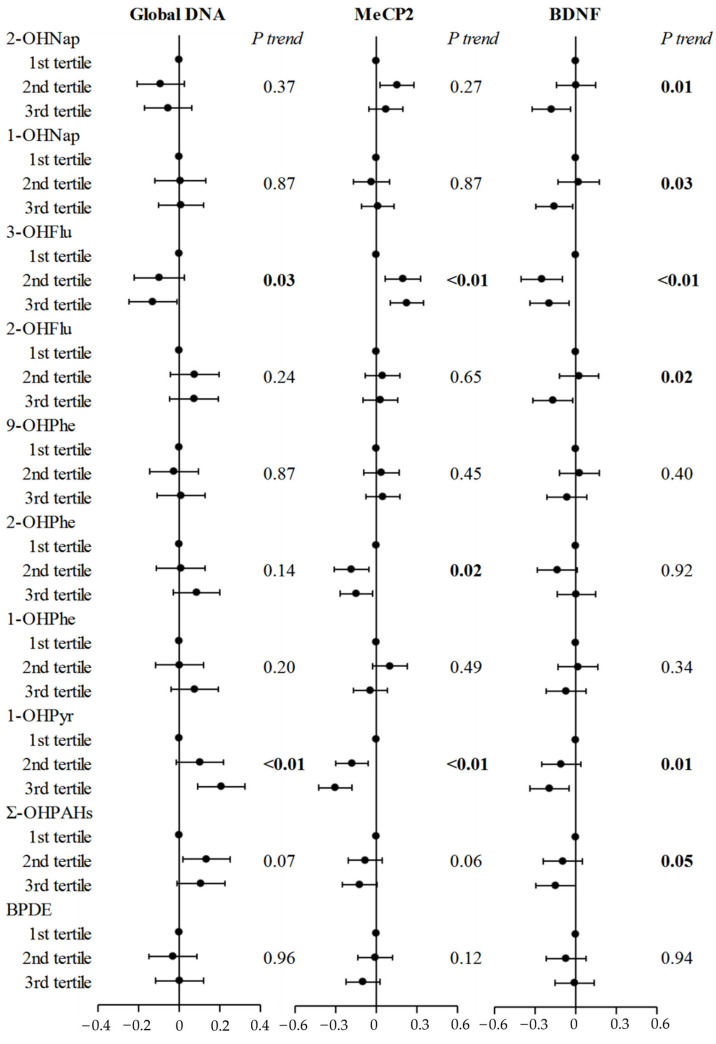
Regression coefficients for the cord blood DNA 5-hmC level associated with PAH metabolites levels after adjusting for mother’s age, mother’s education, pre-pregnant BMI, household income, parity, passive smoking, child’s sex, gestational age, and cord blood lead. Note: Ʃ-OHPAH: sum of 11 individual PAH metabolites; BPDE: (+)benzo[a]pyrene-7,8-dihydrodiol-9,10-epoxide; *MeCP2*: methyl CpG binding protein 2; *BDNF*: brain-derived neurotrophic factor; BMI: body mass index.

**Figure 4 toxics-13-00726-f004:**
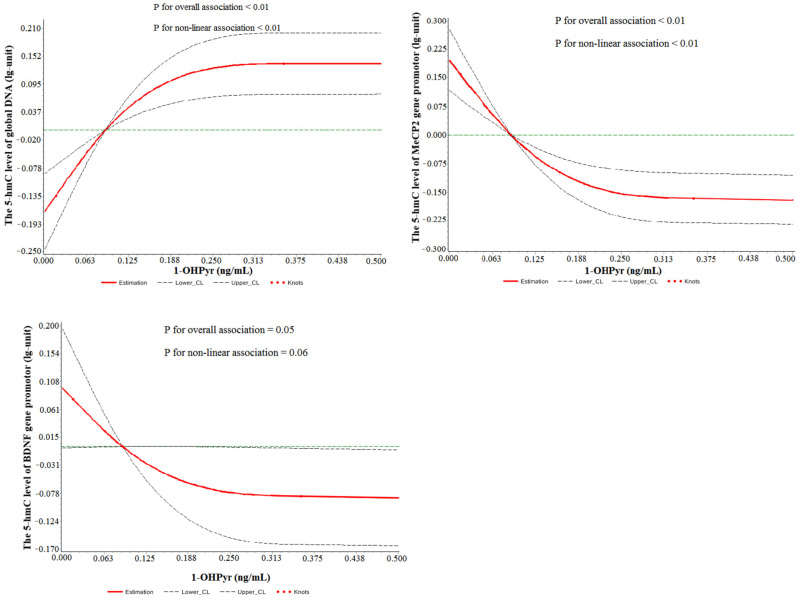
Restricted cubic spline models representing the associations between maternal urinary 1-OHPyr and global DNA 5-hmC and *MeCP2* and *BDNF* gene promoter 5-hmC, adjusted for mother’s age, mother’s education, pre-pregnant BMI, household income, parity, passive smoking, child’s sex, gestational age, and cord blood lead. The reference group is 0.090 ng/mL; dashed lines represent the 95% CI; the green line represent the reference line with a vertical axis of 0; the red knots represent 1-OHPyr concentrations at the 10th, 50th, and 90th percentiles, respectively. Note: *MeCP2*: methyl CpG binding protein 2; *BDNF*: brain-derived neurotrophic factor; BMI: body mass index.

**Table 1 toxics-13-00726-t001:** Characteristics of subjects in the parent study and in this study.

Characteristics	Participants ^a^ (*n* = 221)	Non-Participants (*n* = 179)	*p*-Value
Mother			
Age (years)	28.0 ± 3.8	27.3 ± 4.9	0.114
Pre-pregnancy BMI (kg/m^2^)	21.6 ± 3.2	21.3 ± 2.9	0.311
Educational status			<0.001
Middle school and below	51 (23.1%)	69 (38.5%)	
High school	45 (20.4%)	50 (27.9%)	
College and above	125 (56.6%)	60 (33.6%)	
Household income (CNY/year)			0.048
<30,000	88 (39.8%)	89 (49.7%)	
30,000–50,000	85 (38.5%)	66 (36.9%)	
≥50,000	48 (21.7%)	24 (13.4%)	
Parity (*n* = 1)	155 (70.1%)	123 (68.7%)	0.759
Passive smoking (yes)	102 (46.2%)	96 (53.6%)	0.102
Child			
Gender (male)	116 (52.5%)	97 (54.2%)	0.735
Gestational age (days)	280.9 ± 7.8	279.5 ± 8.6	0.082
Birth weight (g)	3469.5 ± 438.7	3363.9 ± 433.0	0.018
Birth length (cm)	50.7 ± 2.1	50.0 ± 2.2	<0.001
Birth head circumference (cm)	33.9 ± 1.8	33.8 ± 1.8	0.684
NBNA scores	38.7 ± 1.1	38.6 ± 1.1	0.909
Motor	111.6 ± 15.0	-	-
Adaptive	110.5 ± 14.6	-	-
Language	108.8 ± 17.2	-	-
Social	111.7 ± 14.5	-	-

BMI: body mass index; NBNA: neonatal behavioral neurological assessment; continual variable presents mean ± SD; categorical variable presents frequency (proportion). ^a^ Subjects have available follow-up data.

**Table 2 toxics-13-00726-t002:** Distribution of maternal PAH metabolites and cord blood BPDE (*n* = 221).

PAHs	Percent Detection (%)	GM	P25	P50	P75
2-OHNap	98.6%	0.063	0.015	0.072	0.230
1-OHNap	55.2%	0.009	<LOD	0.010	0.053
3-OHFlu	52.5%	0.009	<LOD	0.006	0.070
2-OHFlu	76.0%	0.036	0.003	0.077	0.212
9-OHPhe	93.2%	0.045	0.024	0.057	0.120
2-OHPhe	91.9%	0.078	0.046	0.133	0.194
1-OHPhe	67.4%	0.017	<LOD	0.024	0.131
1-OHPyr	96.4%	0.080	0.038	0.090	0.199
3-OHChr	14.9%	0.007	<LOD	<LOD	<LOD
6-OHChr	18.6%	0.006	<LOD	<LOD	<LOD
9-OHBap	33.5%	0.006	<LOD	<LOD	0.137
∑-OHPAHs	100.0%	0.754	0.343	0.687	1.724
BPDE	95.0%	4.3	3.4	5.2	8.0

LOD: limit of detection; GM: geometric mean; Ʃ-OHPAH: sum of 11 individual PAH metabolites; P25: 25th percentile; P50: 50th percentile (median); P75: 75th percentile.

**Table 3 toxics-13-00726-t003:** Associations between 5-hmC and neurodevelopmental indexes.

5-hmC	β (95% CI)				
	NBNA	Motor	Adaptive	Language	Social
Global DNA	−0.48	−10.66	−14.18	−14.42	−10.26
	(−0.89, −0.08)	(−15.98, −5.34)	(−19.11, −9.25)	(−20.36, −8.47)	(−15.23, −5.28)
*MeCP2*	0.41	8.89	10.03	11.29	8.93
	(0.03, 0.80)	(3.26, 13.33)	(5.27, 14.80)	(5.64, 16.95)	(4.25, 13.61)
*BDNF*	0.10	7.77	0.14	−0.15	1.51
	(−0.23, 0.44)	(3.45, 12.09)	(−4.13, 4.41)	(−5.19, 4.90)	(−2.65, 5.67)

Covariates include mother’s age, mother’s education, pre-pregnant BMI, household income, parity, passive smoking, child’s gender, gestational age, and cord blood lead. The Gesell developmental scale scores additionally adjust the level of PAH metabolites at age 2.

**Table 4 toxics-13-00726-t004:** Mediating effects of 5-hmC on the associations of log-transformed levels of urinary 1-OHPyr with neurodevelopmental scores.

	Exposure to Mediator (β_1-OHPyr_)	Mediator to Outcome (λ_M_)	Mediated Effect (Indirect Effect, β_1-OHPyr_ × λ_M_)	Direct effect (λ_1-OHPyr_)	Mediated Proportion (%)
Global DNA					
Motor	0.15	−9.10	−1.34	−3.36	28.51
	(0.07, 0.22)	(−14.57, −3.63)	(−2.63, −0.48)	(−6.43, −0.28)	
Adaptive	0.15	−13.13	−1.93	−2.26	46.06
	(0.07, 0.22)	(−18.22, −8.04)	(−3.16, −1.01)	(−5.12, 0.60)	
Language	0.15	−13.49	−1.98	−1.99	49.87
	(0.07, 0.22)	(−19.65, −7.33)	(−3.48, −0.90)	(−5.45, 1.47)	
Social	0.15	−9.62	−1.41	−1.37	50.72
	(0.07, 0.22)	(−14.78, −4.47)	(−2.66, −0.61)	(−4.26, 1.53)	
*MeCP2*					
Motor	−0.20	6.41	−1.28	−3.41	27.29
	(−0.28, −0.12)	(1.11, 11.70)	(−2.36, −0.34)	(−6.60, −0.23)	
Adaptive	−0.20	8.68	−1.74	−2.46	41.43
	(−0.28, −0.12)	(3.65, 13.71)	(−2.86, −0.77)	(−5.48, 0.56)	
Language	−0.20	10.23	−2.04	−1.93	51.39
	(−0.28, −0.12)	(4.24, 16.22)	(−3.71, −0.75)	(−5.53, 1.67)	
Social	−0.20	8.31	−1.66	−1.12	40.29
	(−0.28, −0.12)	(3.35, 13.27)	(−2.81, −0.72)	(−4.10, 1.86)	
*BDNF*					
Motor	−0.13	6.74	−0.89	−3.80	18.98
	(−0.23, −0.04)	(2.40, 11.08)	(−2.20, −0.20)	(−6.84, −0.77)	

Covariates include mother’s age, mother’s education, pre-pregnant BMI, household income, parity, passive smoking, child’s gender, gestational age, cord blood lead, and the level of PAH metabolites at age 2.

## Data Availability

The datasets used and/or analyzed during the current study are available from the corresponding author on reasonable request.

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
