# Peer review of "Association of Prenatal Polycyclic Aromatic Hydrocarbons Exposure, DNA Hydroxymethylation, and Neurodevelopment at 0 and 2 Years of Age"

_toxics, 2025, doi:10.3390/toxics13090726_

Round 1

Reviewer 1 Report

Comments and Suggestions for Authors

The presented manuscript informs about the associations between exposure to polycyclic hydrocarbons (PAHs)_ and the 5-hmC changes in the brain-derived neurotrophic factor (BDNF) gene and motor scores in 2-year-old children.  The paper postulates that the 5-hmC change in the neurodevelopmental gene BDNF may be one of the mechanisms by which PAH causes neurodevelopmental toxicity in pregnancy.

The study is the first report of the relationship among maternal exposure to PAH, the levels of global DNA 5-hmC and the specific gene pro-303 moter 5-hmC in cord blood and offspring neurodevelopmental scores at birth and 2 years old.

In this study, maternal urinary 1-OHPyr was associated with lower neurodevelopmental scores in children ages 0-2 years. A significant inverse association between 1 1-OHPyr and the 5-hmC levels of the BDNF and MeCP2 gene promoters was also observed. 1-OHPyr was positively associated with global DNA 5-hmC levels. A mediation analysis indicated that 5-hmC served a mediating role between 1-OHPyr and motor scores. This suggests that PAH exert adverse neurodevelopmental effects through changes in DNA hydroxymethylation modification, and that 5-hmC can serve as an early predictive biomarker for neurodevelopment.

Main remark:

The authors suggest that PAH exert adverse neurodevelopmental effects through changes in DNA hydroxymethylation modification and that 5-hmC can serve as an early predictive biomarker for neurodevelopment. However, no direct relationship was found between PAH prenatal exposure and neurodevelopment.  In the discussion, the authors might inform us of the reason for this.

Minor remarks

Line 88: The sentence „ f pregnant women who...” needs to be corrected.

Line  160; The authors inform „ When newborns were three days old, newborns were administered the NBNA test „ – Is the years old index a proper one? . On the other hand, 221 mother-child pairs with 2-year-old children were included in the study.

Line 211 Table 1. „Characteristics of subjects in the Parent study and this study.”  As the second is part of the first, no statistical comparison is possible. The compared groups should be participants vs nonparticipants.  

Line 223 Table 2. Distribution of maternal PAH metabolites and cord blood BPDE (n=221).  What are the abbreviations GM and P25? P50. P75  is in the text above the table. The median levels are presented.

Author Response

Dear editor,

On behalf of all the contributing authors, I would like to express our article entitled “Association of Prenatal Polycyclic Aromatic Hydrocarbons Exposure, DNA Hydroxymethylation, and Neurodevelopment at 0 and 2 Years of Age” (Manuscript No. 3766688). These comments are valuable and helpful for improving our article. According to peer review report and editor’s comments, we have made extensive discussions and analyses based on relevant researches to clarify the clinical value of our study. And all changes made in the revised manuscript have trace.

Thank you again for your positive comments and valuable suggestions to improve the quality of our manuscript. I hope you will find this revised manuscript acceptable for publication in Toxics.

Sincerely,

Jisheng Nie

Department of Occupational and Environmental Health, School of Public Health, Shanxi Medical University, Xinjiannan Road 56, Taiyuan 030001, China

Email: niejisheng@sxmu.edu.cn

Response to Reviewer:

Reviewer #1:

The presented manuscript informs about the associations between exposure to polycyclic hydrocarbons (PAHs) and the 5-hmC changes in the brain-derived neurotrophic factor (BDNF) gene and motor scores in 2-year-old children.  The paper postulates that the 5-hmC change in the neurodevelopmental gene BDNF may be one of the mechanisms by which PAH causes neurodevelopmental toxicity in pregnancy.

The study is the first report of the relationship among maternal exposure to PAH, the levels of global DNA 5-hmC and the specific gene pro-303 moter 5-hmC in cord blood and offspring neurodevelopmental scores at birth and 2 years old.

In this study, maternal urinary 1-OHPyr was associated with lower neurodevelopmental scores in children ages 0-2 years. A significant inverse association between 1 1-OHPyr and the 5-hmC levels of the BDNF and MeCP2 gene promoters was also observed. 1-OHPyr was positively associated with global DNA 5-hmC levels. A mediation analysis indicated that 5-hmC served a mediating role between 1-OHPyr and motor scores. This suggests that PAH exert adverse neurodevelopmental effects through changes in DNA hydroxymethylation modification, and that 5-hmC can serve as an early predictive biomarker for neurodevelopment.

Main remark:

Comment 1. The authors suggest that PAH exert adverse neurodevelopmental effects through changes in DNA hydroxymethylation modification and that 5-hmC can serve as an early predictive biomarker for neurodevelopment. However, no direct relationship was found between PAH prenatal exposure and neurodevelopment.  In the discussion, the authors might inform us of the reason for this.

Response 1. Thank you for your valuable suggestions. In this study, the adverse effects of PAH on neurodevelopment were mainly reflected in two aspects. Firstly, PAH had a direct effect on neurodevelopment indicators. Secondly, PAH indirectly affected neurodevelopment indicators by affecting DNA hydroxymethylation modification.

Mediation effects are divided into partial mediation effects and complete mediation effects. Partial mediation effect refers to the presence of both direct and mediating effects, where exposure (X) directly affects outcome (Y) (coefficient λ1-OHPyr is significant), while another portion affects outcome (Y) through mediating variable M (coefficient β 1-OHPyr * λ M is also significant). The complete mediating effect means that exposure (X) cannot directly affect outcome (Y), and must be transmitted through the mediating variable M. At this point, the coefficient is λ1-OHPyr is not significant.

In our study, we found significant mediation effects (indirect effects) between maternal urinary 1-OHPyr and GDS score (including motor, adaptive, language, and social) for global DNA 5-hmC and MeCP2 and BDNF gene promoter 5-hmC (P for mediator < 0.05). About motor score, the global DNA 5-hmC and MeCP2 and BDNF gene promoter 5-hmC could explain 28.51%, 27.29%, and 18.98%, respectively, of the effect of 2-year-old children’s motor scores changes related to 1-OHPyr. The direct effects of 1-OHPyr on the motor scores were 71.49%, 72.71%, and 81.02%, respectively.

But, the direct effect of 1-OHPyr on the adaptive scores, language score, and social score was not significant, indicating that 5-hmC played a complete mediating effect between 1-OHPyr and these three scores (adaptive scores, language score, and social score), reflecting that the impact of 1-OHPyr on adaptive scores, language score, and social score was achieved through its influence on 5-hmC. As for the motor score, the decrease in motor score caused by 1-OHPyr includes both direct and indirect effects.

We have provided a detailed description and explanation of these in the results, please refer to the revised manuscript for details.

Minor remarks

Comment 2. Line 88: The sentence „ f pregnant women who...” needs to be corrected.

Response 2. Thank you for your valuable suggestions. According to the reviewer’s comment, we have made revisions, as detailed in the methodology section of the revised manuscript.

The specific modifications are as follows: “Pregnant women who waited for delivery during the third trimester of pregnancy (≥ 35 weeks) in the Sixth Hospital of Shanxi Medical University and the Eighth People’s Hospital of Taiyuan were invited to participate in the Taiyuan Mother and Child Cohort Study. Eligibility criteria included the followings: eligible women had resided in Taiyuan city for at least 1 year, were ≥18 years old, were nonsmokers, had no family history of neurological diseases and chronic diseases, and had a single gestational viable fetus. Before entering the study, informed consent was obtained from mothers. If pregnant women met the inclusion criteria and signed informed consent, we collected maternal urine before delivery. Umbilical cord blood was collected after delivery. A total of 400 mother-newborn pairs were included in the cohort study. The present study obtained approval from the Ethical Committee of Shanxi Medical University Institutional Review Board (No. 2016LL087). Two years later, 221 mother-child pairs were included in the cohort.”

Comment 3. Line  160; The authors inform „ When newborns were three days old, newborns were administered the NBNA test „ – Is the years old index a proper one? . On the other hand, 221 mother-child pairs with 2-year-old children were included in the study.

Response 3. Thank you for your valuable suggestions. NBNA is a scale used to assess the neurobehavioral development of newborns [1-2]. The scale contained five clusters: behavior (6 items); passive tone (4 items); active tone (4 items); primary reflexes (3 items); and general assessment (3 items). Each item had three scales (0, 1, and 2). Twenty items had a maximum of 40 scores. The assessment is primarily conducted by neonatologists within the first week after birth to reflect the newborn's behavior. This scale is commonly used in clinical settings, and specific methods can be found in our previous studies[3].

A total of 221 cases were used for NBNA analysis in this study. This cohort primarily recruited 400 mother-child pairs as participants, but only 221 mother-child pairs completed the 2-year follow-up. To maintain consistency between the two outcomes, the sample size for this study was 221 cases.

Reference

[1] Bao, X.L., et al., 1991. Twenty-item behavioral neurological assessment for normal newborns in 12 cities of China. Chin Med J (Engl). 104, 742–746.

[2] Bao, X.L., et al., 1993. 20-item neonatal behavioral neurological assessment used in predicting prognosis of asphyxiated newborn. Chin Med J (Engl). 106, 211–215.

[3] Nie J, ., et al., 2019. Prenatal polycyclic aromatic hydrocarbons metabolites, cord blood telomere length, and neonatal neurobehavioral development. Environ Res. 174:105-113.

Comment 4. Line 211 Table 1. „Characteristics of subjects in the Parent study and this study.”  As the second is part of the first, no statistical comparison is possible. The compared groups should be participants vs nonparticipants.  

Response 4. Thank you for your valuable suggestions. We have reorganized Table 1, comparing baseline characteristics between participants and non-participants according to reviewer comments, as detailed in the revised manuscript Table 1.

Comment 5. Line 223 Table 2. Distribution of maternal PAH metabolites and cord blood BPDE (n=221).  What are the abbreviations GM and P25? P50. P75  is in the text above the table. The median levels are presented.

Response 5. Thank you for your valuable suggestions. Table 2 presents the distribution of maternal PAH metabolites and cord blood BPDE levels, where GM stands for geometric mean, used to describe the central tendency of skewed data [1]. The purpose of providing the geometric mean in Table 2 is to facilitate comparisons with similar studies. The P25, P50, and P75 represent the 25th percentile, median (50th percentile), and 75th percentile, respectively. Geometric mean and median are both indicators that reflect the central tendency of skewed data. In order to provide more information when comparing similar studies, we have provided both geometric mean and median in Table 2. Based on reviewer comments, we added annotations to clarify the abbreviations.

Reference

  • Vogel, Richard M.. “The geometric mean?” Communications in Statistics - Theory and Methods 51 (2020): 82 - 94.

We would like to take this opportunity to thank you for all your time involved and this great opportunity for us to improve the manuscript. We hope you will find this revised version satisfactory.

Sincerely,

Jisheng Nie

Reviewer 2 Report

Comments and Suggestions for Authors

This is an interesting paper that explores .the association between prenatal exposure to PAHs, DNA hydroxymethylation (5-hmC), and neurodevelopmental outcomes in children at birth and before 2 years of age.   Before the paper can be published some minor corrections should be done.

  • There are several typos:  Line 13: “may be associate with”  should be "may be associated with";
  • Line 40: “Epigenome change often serve as to discover biomarkers” should be "Epigenome changes often serve to discover biomarkers".
  • Line 114: “HPLC-MS/MS (Shimadzu, Kyoto, Japan))” Double closing parenthesis
  • Line 179: “were less than the limit of detection was given a value” → should be “were less than the limit of detection were given a value”.
  • No information is provided on whether other environmental confounders (e.g., heavy metals or co-pollutants) were adjusted for beyond lead.
  • Comment in the discussion that a single time point for PAH exposure may not capture the variability across pregnancy.
  • The possibility of reverse causality is not discussed (e.g., whether fetal characteristics could influence maternal metabolism or measurement).
  • The mediation analysis lacks discussion on assumptions (e.g., temporal precedence and no unmeasured confounding).
  • The criteria for excluding PAH metabolites with <50% detection is mentioned but not fully justified in terms of statistical implications.
  • It is not clear whether the ELISA kits used for 5-hmC quantification were validated in cord blood specifically.
  • There is no mention of batch effects for HPLC-MS/MS, nor whether lab technicians were blinded to case/control status.
  • The method for handling values below LOD (assigning ½ LOD) is mentioned but include a  justification or a sensitivity analysis.
  • More details about qPCR primer sequences (or a citation to a supplemental table) would help reproduce the immunoprecipitation assays.
  • Give a short comment on effect sizes in practical terms (e.g., what a 10-point change in motor score means clinically).
  • More discussion is needed on why only 1-OHPyr consistently showed associations while others did not was this due to detection levels or unique properties?
  • Explain whether interaction terms (e.g., sex-specific effects) were tested or explored.
  • Explain the basis for choosing 1-OHPyr as the main exposure of interest in Figures 1–4 (e.g., why not ∑OHPAHs?).
  • It is speculative in some parts (e.g., ROS effects on TET activity) without citing direct evidence from human studies.
  • The possibility of residual confounding (e.g., nutrition, genetic polymorphisms) is not discussed.
  • The issue of sample attrition and potential selection bias (higher education in follow-up group) needs more emphasis.
  • The discussion could better highlight policy or public health implications (e.g., screening for 5-hmC in cord blood?).

Author Response

Dear editor,

On behalf of all the contributing authors, I would like to express our article entitled “Association of Prenatal Polycyclic Aromatic Hydrocarbons Exposure, DNA Hydroxymethylation, and Neurodevelopment at 0 and 2 Years of Age” (Manuscript No. 3766688). These comments are valuable and helpful for improving our article. According to peer review report and editor’s comments, we have made extensive discussions and analyses based on relevant researches to clarify the clinical value of our study. And all changes made in the revised manuscript have trace.

Thank you again for your positive comments and valuable suggestions to improve the quality of our manuscript. I hope you will find this revised manuscript acceptable for publication in Toxics.

Sincerely,

Jisheng Nie

Department of Occupational and Environmental Health, School of Public Health, Shanxi Medical University, Xinjiannan Road 56, Taiyuan 030001, China

Email: niejisheng@sxmu.edu.cn

Response to Reviewer:

Reviewer #2:

This is an interesting paper that explores .the association between prenatal exposure to PAHs, DNA hydroxymethylation (5-hmC), and neurodevelopmental outcomes in children at birth and before 2 years of age.   Before the paper can be published some minor corrections should be done.

Comment 1. There are several typos:  Line 13: “may be associate with”  should be "may be associated with";

Line 40: “Epigenome change often serve as to discover biomarkers” should be "Epigenome changes often serve to discover biomarkers".

Line 114: “HPLC-MS/MS (Shimadzu, Kyoto, Japan))” Double closing parenthesis

Line 179: “were less than the limit of detection was given a value” → should be “were less than the limit of detection were given a value”.

Response 1. Thank you for your reminder. We have made corrections and conducted a detailed inspection.

Comment 2. No information is provided on whether other environmental confounders (e.g., heavy metals or co-pollutants) were adjusted for beyond lead.

Response 2. Thank you for your valuable comment. We may not have clearly explained the specific confounders in manuscript. Based on the reviewer’s comment, we further detailed the confounders in revised manuscript. In this study, we only adjusted for the confounding effects of lead. Other heavy metals and co-pollutants mentioned by the reviewer were not detected and analyzed in this study. This is a possible research topic for our future.

Comment 3. Comment in the discussion that a single time point for PAH exposure may not capture the variability across pregnancy.

Response 3. Thank you for your valuable comment. We must acknowledge an important limitation of this study: we know that pregnancy spans roughly 40 weeks. The PAH hydroxymetabolites, serving as indicators of internal PAH exposure, can only indicate recent exposure levels. During the research design phase, sampling should be performed at various stages of pregnancy, encompassing early, middle, and late stages, to ascertain the critical window for neurotoxicity exposure and to accurately estimate the cumulative PAH exposure dose throughout pregnancy. This study, however, only collected urine samples from late pregnancy, based on two factors: (1) previous reports indicating that late pregnancy is a critical period for neurodevelopment [1-2]; (2) the convenience and feasibility of sample collection, given the significant changes in prenatal examination hospitals during early and middle pregnancy, which complicate the process of follow-up.

Reference

[1] Laura G, Stefano P, Saal FS, Vom, Paola P: The effects of bisphenol A on emotional behavior depend upon the timing of exposure, age and gender in mice. Hormones & Behavior 2013, 63(4):598-605.

[2] Meredith RM: Sensitive and critical periods during neurotypical and aberrant neurodevelopment: a framework for neurodevelopmental disorders. Neuroscience and biobehavioral reviews 2015, 50:180-188.

Comment 4. The possibility of reverse causality is not discussed (e.g., whether fetal characteristics could influence maternal metabolism or measurement).

Response 4. Thank you for your valuable comment. We indeed overlooked this aspect of the discussion in the original manuscript. We have add the discussion on impact of fetal characteristics on maternal PAH metabolism or measurement in the revised manuscript.

We think that reverse causality argument does not hold in this article, as causality has temporal logic. We explore the effects of prenatal exposure on offspring neuro development, particularly at the age of 2, where PAH exposure occurred before the outcome.

The potential impact of fetal characteristics on PAH metabolism, as mentioned by the reviewer, may indeed exist, which reflects factors affecting maternal metabolism, not the focus of our study in this article. PAH metabolites reflect the level of PAH exposure during pregnancy as internal exposure. If the fetus influences metabolism, it also affects all pregnant women. The relative concentration of PAH metabolites remains consistent and does not affect the evaluation of exposure levels. What may be affected is solely the absolute amount of internal exposure.

Comment 5. The mediation analysis lacks discussion on assumptions (e.g., temporal precedence and no unmeasured confounding).

Response 5. Thank you for your valuable comment. In the methodology section, we provided the function and applicable conditions of mediation analysis, and in Results 3.6, we also explained why mediation analysis was conducted in this study. In response to the reviewer's comments, we have added the prerequisites for mediation analysis and unmeasured confounding in the discussion section of the revised manuscript.

Comment 6. The criteria for excluding PAH metabolites with <50% detection is mentioned but not fully justified in terms of statistical implications.

Response 6. Thank you for your valuable comment. To ensure statistical efficiency, sufficient sample size is a prerequisite for analyzing the relationship between PAH metabolites and outcomes. Therefore, PAH metabolites with a detection rate of less than 50% were excluded from the analysis in this study. In addition, to eliminate the influence of these excluded metabolites on the results, we analyzed metabolites with a detection rate of less than 50% as covariates in the sensitivity analysis and found that it did not change the original conclusion (Table S2). We have discussed and analyzed the potential impacts and results of handling metabolites with detection rates < 50%, as detailed in the discussion section of the revised manuscript.

Comment 7. It is not clear whether the ELISA kits used for 5-hmC quantification were validated in cord blood specifically.

Response 7. The sample in this study was DNA from umbilical cord blood. First, umbilical cord blood DNA was extracted, and then DNA samples that met the requirements were tested. During the experiment, we did not find any differences in concentration and purity between umbilical cord blood DNA and peripheral blood DNA. Although the instructions for ELISA kit do not specify its specificity for umbilical cord blood, this kit has shown good sensitivity and specificity in detecting the DNA hydroxymethylation marker 5-hmC and has been widely used in scientific research experiments [1-2].

Reference

[1] Sanchez-Guerra M, Zheng Y, Osorio-Yanez C, Zhong J, Chervona Y, Wang S, Chang D, McCracken JP, Díaz A, Bertazzi PA et al: Effects of particulate matter exposure on blood 5-hydroxymethylation: results from the Beijing truck driver air pollution study. Epigenetics 2015, 10(7):633-642

[2] Yang C et. al. (February 2025). The Diminution of R-Loops Generated by LncRNA DSP-AS1 Inhibits DSP Gene Transcription to Impede the Re-Epithelialization During Diabetic Wound Healing. Adv Sci (Weinh).:e2406021.

Comment 8. There is no mention of batch effects for HPLC-MS/MS, nor whether lab technicians were blinded to case/control status.

Response 8. Thank you for your valuable suggestions. In order to eliminate batch effects for HPLC-MS/MS, we applied the following treatment: firstly, we re made the working curve every day during detection; Secondly, each version of the sample has three different concentration gradient quality control points; Thirdly, the order of sample detection is random. The linear relationship of the work curve in this study is good, and the linear correlation coefficients are all greater than 0.99; The relative standard deviation (RSD) reflecting precision is 2.7% -11.6%; The recovery rate of spiked PAHs was 71.4% -109.4%. For the methodology of HPLC detection of PAHs metabolites, please refer to our previous report [1]. The testing personnel were laboratory technicians who were unaware of the grouping of samples. All samples were loaded onto the machine with uniform numbering without group markings.

Reference

[1] Cheng Lin, Li Jinyu, Lv Shengjie, Niejisheng. Detection of polycyclic aromatic hydrocarbon hydroxyl metabolites in urine by ultra-high performance liquid chromatography tandem mass spectrometry [J]. Environmental and Occupational Medicine, 2017, 34 (11): 1004-1008. DOI: 10.13213/j.cnki. jeom. 2017.17317.

Comment 9. The method for handling values below LOD (assigning ½ LOD) is mentioned but include a  justification or a sensitivity analysis.

Response 9. Thank you for your valuable comment. For samples with a detection rate of less than 20% metabolites and below LOD, 1/2 of LOD can be used to replace undetected samples in statistical analysis. This is a common statistical processing method in environmental toxicology [1-3]. This processing can maximize the reflection of the information of the detection sample. The traditional direct exclusion method will lose the information of this part of the data, especially when designing grouping, this processing method will not affect the results. In this study, PAH metabolites were mainly involved in association analysis as categorical variables, therefore, the main results of this study were not affected by this treatment.

Reference

  • Jacob P 3rd, Wilson M, Benowitz NL. Determination of phenolic metabolites of polycyclic aromatic hydrocarbons in human urine as their pentafluorobenzyl ether derivatives using liquid chromatography-tandem mass spectrometry. Anal Chem. 2007 Jan 15;79(2):587-98.
  • Du J, Pan B, Cao X, Li J, Yang J, Nie J. Urinary polycyclic aromatic hydrocarbon metabolites, peripheral blood mitochondrial DNA copy number, and neurobehavioral function in coke oven workers. Chemosphere. 2020 Dec;261:127628. doi: 10.1016/j.chemosphere.2020.127628.
  • Cao X, Li J, Cheng L, Deng Y, Li Y, Yan Z, Duan L, Yang J, Niu Q, Perera F, Nie J, Tang D. The associations between prenatal exposure to polycyclic aromatic hydrocarbon metabolites, umbilical cord blood mitochondrial DNA copy number, and children's neurobehavioral development. Environ Pollut. 2020 Oct;265(Pt B):114594. doi: 10.1016/j.envpol.2020.114594.

Comment 10. More details about qPCR primer sequences (or a citation to a supplemental table) would help reproduce the immunoprecipitation assays.

Response 10. Based on the reviewers' suggestions, we have added the primer sequences in Supplementary material Table S1.

Table S2. Primer sequences of BDNF and MeCP2 genes

Gene symbol

Chromosomal location

Primer sequence (5 '-3')

BDNF

chr11:27654893-27720779

F: CCCACCCACTTTCCCATTCA

R: CGGAGGTAATACTCGCACCC

MeCP2

chrX:154021573-154097755

F: GCCCACTAAACCAGTCCCTC

R: ACCCCTCCAGCTGTTGATTG

Comment 11.Give a short comment on effect sizes in practical terms (e.g., what a 10-point change in motor score means clinically).

Response 11. Thank you for your valuable comment. We have added the significance and value of PAH effects in practical terms in the result and discussion section of the revised manuscript.

“The motor score mainly reflected the development of gross and fine motor skills in children. Low motor scores were associated with a higher risk of developing autism spectrum disorder, while 1-OHPyr was negatively correlated with motor scores. Compared to the 1st tertile, the 3rd tertile group showed an decrease of 10.3 points in motor scores and a decrease of 9.2% compared to the average level, indicating an increased risk of developing autism and motor delay.”

Comment 12. More discussion is needed on why only 1-OHPyr consistently showed associations while others did not was this due to detection levels or unique properties?

Response 12. Thank you for your valuable comment. We have analyzed the reasons for the consistent associations between 1-OHPyr and neurodevelopmental indicators in the discussion section of the revised manuscript.

Comment 13. Explain whether interaction terms (e.g., sex-specific effects) were tested or explored.

Response 13. At the beginning of the analysis, we set the interaction terms, but there was no statistical significance, so we only considered the main effect. At the same time, in order to control for the possible influence of gender, we treated gender as a covariate in multiple regression analysis.

Comment 14. Explain the basis for choosing 1-OHPyr as the main exposure of interest in Figures 1–4 (e.g., why not ∑OHPAHs?).

Response 14. Thank you for your valuable suggestions. In Figure 1, we conducted preliminary analysis on various PAH metabolites and found that only 1-OHPyr was associated with multiple indicators of neurodevelopment, including Motor, Adaptive, Language and Social. This indicates that 1-OHPyr is a risk factor affecting neurodevelopment. To confirm this relationship, we further visualized the dose-response relationship between the two using the RCS model. Similarly, when exploring the correlation between PAH and 5-hmC (Figure 3), we also found the role of 1-OHPyr, and further analyzed the dose-response relationship in Figure 4. The premise of mediation analysis is: firstly, exposure is related to the outcome; Secondly, exposure is related to mesons; The third meson is related to the outcome. Based on the previous results, we found that only 1-OHPyr meets these conditions. Therefore, mediation analysis can only present the overall effect, direct effect, and indirect effect of 1-OHPyr on the outcome.

Comment 15. It is speculative in some parts (e.g., ROS effects on TET activity) without citing direct evidence from human studies.

Response 15. Thank you for your valuable suggestions. We introduced the speculative of ROS on TET enzyme activity in order to further infer the rationality of causal relationship from a biological perspective. Currently, there are no population-based studies reporting the effect of ROS on TET enzyme activity, but there is data from animal model and cell, which indirectly supports the biological rationality of our results.

Comment 16. The possibility of residual confounding (e.g., nutrition, genetic polymorphisms) is not discussed.

Response 16. In the discussion section of the revised manuscript, we added the possible impact of nutrition and genetic polymorphism on the results in limitation section.

Comment 17. The issue of sample attrition and potential selection bias (higher education in follow-up group) needs more emphasis.

Response 17. Thank you for your valuable suggestions. In the discussion section of the revised manuscript, we have added limitations regarding the impact of loss to follow-up on this study.

Comment 18. The discussion could better highlight policy or public health implications (e.g., screening for 5-hmC in cord blood?).

Response 18. Thank you for your valuable suggestion. We have added the value and significance of this study for public health and policy in discussion section of revised manuscript.

We would like to take this opportunity to thank you for all your time involved and this great opportunity for us to improve the manuscript. We hope you will find this revised version satisfactory.

Sincerely,

Jisheng Nie